# Management of veterinary anaesthesia in small animals: A survey of current practice in Quebec

Geoffrey Truchetti[1], Colombe Otis[2], Anne-Claire Brisville[3], Guy Beauchamp[2], Daniel Pang[2], Eric Troncy [2]*

1 Centre Vétérinaire Rive-Sud, Brossard, Québec, Canada, 2 Groupe de Recherche en Pharmacologie Animale du Québec (GREPAQ), Faculty of Veterinary Medicine, Université de Montréal, Saint-Hyacinthe, Quebec, Canada, 3 Boehringer Ingelheim Animal Health, Bovine Unit, Burlington, Ontario, Canada

* eric.troncy@umontreal.ca

## Abstract

### Objective

To describe how small animal anaesthesia is performed in French-speaking Eastern Canada, and the variations between practices, in particular based on practice type, veterinarian gender and experience.

### Design

Observational study, survey.

### Sample

156 respondents.

### Procedure

A questionnaire was designed to assess current small animal anaesthesia practices in French-speaking Eastern Canada, mainly in the province of Quebec. The questionnaire was available through SurveyMonkey, and consisted of four parts: demographic information about the veterinarians surveyed, evaluation and management of anaesthetic risk, anaesthesia procedure, monitoring and safety. Gender, year of graduation, and type of practice were tested as potential risk factors. Chi-square exact test was used to study relations between each risk factor, and the effect of the selected risk factor on each response of the survey. For ordinal data, the Cochran-Mantel-Haenszel test was used to maximize power.

### Results

Response rate over a period of 3 months was 20.85% (156 respondents). Overall, the way anaesthesia is performed by most respondents does not meet international guidelines, such as patient preparation and evaluation prior to anaesthesia, not using individualised protocols (for 41%), not obtaining intravenous access (12.4% use it for all their anaesthesia in cats, and 30.6% in dogs), lack of patient monitoring at certain intervals for 55% of the responses,

**Data Availability Statement:** Data are presented in Supporting Information files, named S1 to S3 Files.

**Funding:** There was not proprietary interest or funding directly provided for this project or to any

of the authors. This work was indirectly supported (ET) by a Discovery grant (#441651–2013, supporting salaries) and a Collaborative Research and Development grant (#RDCPJ 491953–2016 supporting operations and salaries in partnership with ArthroLab Inc.) from the Natural Sciences and Engineering Research Council (NSERC) of Canada. CO is a recipient of a MITACS Canada Elevation postdoctoral scholarship (#IT11643). ACB received support in the form of salary from the Boehringer Ingelheim Animal Health, but she did participate in the survey outside of her regular tasks for the employer, which had in consequence no role in the present study. The authors got support from the company Dispomed Inc., i.e. to deliver the electronic survey to their clients, with the previous approval of the Ordre des médecins vétérinaires du Québec. The specific roles of all authors are articulated in the 'author contributions' section. The funders had no role in study design, data collection and analysis, decision to publish, or preparation of the manuscript.

**Competing interests:** We feel that our research brings new information to the veterinary anaesthesia knowledge and has the potential to modify future endeavours in the process of conducting anaesthesia for veterinary pet patients. The authors have read the journal's policy and the authors of this paper have the following competing interests: ArthroLab Inc., as partner in a funding grant. The authors received support from the company Dispomed Inc., i.e. to deliver the electronic survey to their clients, with the previous approval of the Ordre des médecins vétérinaires du Québec. The funders, either governmental (NSERC and MITACS) or commercial (ArthroLab Inc., Dispomed Inc.) had no role in the present study. ACB is currently employed by Boehringer Ingelheim Animal Health, but she did participate in the survey outside of her regular tasks for the employer, which had in consequence no role in the present study. This does not alter our adherence to PLOS ONE policies on sharing data and materials. There are no patents, products in development or marketed products to declare.

and client prompted optional analgesia (for 29% of respondents). Some practices are more compliant than others. Among them, referral centres generally offer better care than general practices.

## Conclusions and clinical relevance

The level of care in anaesthesia and analgesia in practices in French-speaking Eastern Canada is concerning, highlighting the need for more sustained continuing education.

## Introduction

Small animals anaesthesia is performed nearly daily in most veterinary practices of French-speaking Eastern Canada. Anaesthesia is not without risks, with mortality reported to be 0.1 to 0.3% in healthy dogs and cats, and considerably higher in sick patients [1–10]. Evaluating anaesthetic/peri-anaesthetic care includes American Society of Anesthesiologists (ASA) physical status classification, and morbidity, considering the common adverse effects such as hypotension, hypothermia, hypoxaemia and hypoventilation [2–8, 11]. In order to improve the management of anaesthesia and perioperative analgesia, guidelines have been published, based on best current evidences and what is accepted as best practice [12–16].

Equipment and drug availability, anaesthesia knowledge and proficiency, as well as access to anaesthesia training and professional environment may be sources of variation in anaesthesia management [13, 14]. There is currently no published evidence of how small animal anaesthesia is practiced in Quebec, whereas a previous study showed some geographical heterogeneity between Canadian provinces in analgesia management [17]. The objective of this observational study was to describe how small animal anaesthesia is performed in this Canadian province. Our hypothesis was that standards of practice will vary among practices in small animals and will not necessarily follow published guidelines. We also suspected some influence of gender, experience and type of practice on the level of anaesthetic care.

## Materials and methods

### Questionnaire

Members of the Research Group in Animal Pharmacology of Quebec (GREPAQ) developed a questionnaire (for detailed questions and choice of answers, see S1 Appendix), designed to assess current small animal anaesthesia practices in French-speaking Eastern Canada, mainly in the province of Quebec. The internal content and construct validation included a pilot survey with a focus group. The latter included various degrees of expertise in veterinary anaesthesia, from veterinary student, general practitioner to anaesthetist in private practice and academia. They evaluated and validated all sections as well as all used terminology to be perfectly understood for any registered veterinary general practitioner, which was the expected audience of the survey. The Ethics Committee for Research in Health and Sciences (CERSES) of Université de Montréal confirmed that such quality improvement in veterinary practice study fell under the Article 2.5 of the Tri-Council Policy Statement of Canada; Ethical Conduct of Research Involving Humans, 2nd edition 2014 (http://www.pre.ethics.gc.ca/eng/policy-politique/initiatives/tcps2-eptc2/Default/) of the activities not requiring research ethics board review.

The questionnaire was available through SurveyMonkey via an electronic link that was sent by email, and consisted of four parts. Part I collected demographic information about the veterinarians surveyed. Part II focused on the evaluation and management of small animals anaesthetic risk. Part III investigated the anaesthesia procedure and finally, in Part IV, respondents evaluated the monitoring and safety of anaesthesia, including during the post-anaesthetic period. Response rate over a period of 3 months, March to May, 2016 was 20.85% (156 respondents) in Quebec small animal practitioners (748 sent invitations).

## Statistical analysis

An independent observer (COT) validated the data by first manually checking records from the SurveyMonkey report and, second editing the descriptive statistics. For inferential statistical analysis, the selected demographic characteristics described in Part I, namely Gender, year of graduation, and type of practice, were tested as potential risk factors influencing the responses in the following sections. Chi-square exact test was used to study relations between each risk factor, and the effect of the selected risk factor on each response of the survey. For ordinal data, the Cochran-Mantel-Haenszel test was used to maximize power. Alpha threshold of 5% was applied for a two-sided analysis. Not all participants responded to all questions. Therefore, descriptive statistical results are expressed in percentage with the ratio of the exact number of answers on the total respondents for each question. Statistical analyses were performed with SAS v.9.3 (SAS Institute, Cary, NC, USA) and results are showed in percentage of the significant risk factor direction effect on each answers and statistical $P$ associated for the statistically significant difference.

## Results

### Part I—Demographic data

A total of seven (7) demographic characteristics of French-speaking respondents are presented (Table 1) with the distribution of each risk factor.

**Risk factors.** Significant associations observed between risk factors are summarised (Table 2). There was a gender effect in year since graduation ($P = 0.01$): 80% of men and 49% of women in respondents graduated more than 15 years ago. There was a significant association between gender and emergency duty ($P = 0.002$) with an overrepresentation of men (59%) compared to women (32%). There was no significant association between gender and the type of practice ($P = 0.83$), the number of veterinarian(s) in the practice ($P = 0.67$), and the number of animals anaesthetised per day ($P = 0.06$).

There was a significant association between years since graduation and the number of veterinarian(s) in the practice ($P = 0.003$), the type of practice ($P = 0.001$) and the number of animals anaesthetised per day ($P < 0.001$). Also, respondents graduated less than 15 years ago more often work in large referral centre ($P = 0.001$), with several veterinarians ($P = 0.003$) doing a lot of anaesthesia cases per day ($P < 0.001$). There was no significant association between years since graduation and emergency duty ($P = 0.51$).

Finally, more animals are anaesthetised in referral centres ($P < 0.001$) or practices with more veterinarians ($P < 0.001$). Unless stated otherwise, the demographic characteristics did not have any influence on the subsequent responses.

### Part II—Evaluation and management of anaesthetic risk

**Client management.** Among respondents, 55% (83/150) provide pamphlet or other information material explaining anaesthesia procedure and related risk. Respondents in referral

**Table 1. Demographic characteristics of 156 veterinarians responding to a survey on management of anaesthesia in small animal practices in French-speaking Canada.**

| Characteristic | | Distribution |
|---|---:|---|
| Gender | | |
| | Male | 44/156 (28.2%) |
| | Female | 112/156 (71.8%) |
| Year of graduation | | |
| | <15 years ago | 76/156 (48.7%) |
| | >15 years ago | 80/156 (51.3%) |
| Number of veterinarian(s) in the practice | | |
| | 1 | 16/156 (10.3%) |
| | 2–4 | 69/156 (44.2%) |
| | 5+ | 71/156 (45.5%) |
| On-call hours | | |
| | Yes | 29/156 (18.6%) |
| | No | 95/156 (60.9%) |
| | Episodic | 32/156 (20.5%) |
| Size of town (population) | | |
| | Very large city (>100 000) | 59/156 (37.8%) |
| | Large city (50 000 to 100 000) | 29/156 (18.6%) |
| | Middle-size town (10 000 to 50 000) | 42/156 (26.9%) |
| | Small town (<10 000) | 26/156 (16.7%) |
| Type of practice | | |
| | General practice (GP) | 124/156 (79.5%) |
| | Referral centre | 32/156 (20.5%) |
| Number of animal(s) anaesthetised/day | | |
| | 0–1 | 17/156 (10.9%) |
| | 2–3 | 41/156 (26.3%) |
| | 4–6 | 50/156 (32.0%) |
| | 7–9 | 21/156 (13.5%) |
| | 10+ | 27/156 (17.3%) |

centre are less likely to use information material (11% *vs*. 89%, *P* = 0.02) than respondents in general practices (GP). Twenty-nine percent (29%, 44/150) of respondents offer analgesia protocol as optional, none of them work in referral centre (so 37%, 44/119, in GP).

An informed consent form is provided to and signed by the owner in 92% (134/146) of the practices. Respondents graduated less than 15 years ago use more often an informed consent form than respondents graduated more than 15 years ago (97% *vs*. 87%, *P* = 0.03).

**Pre-anaesthetic fasting.** Ninety-eight percent (98%, 139/142) of respondents fast healthy patients for 6 to 12 hours prior to anaesthesia in small animals. Fifty-one percent (51%, 71/139) of respondents give free access to water to healthy patients before anaesthesia.

Twelve percent (12%, 17/141) of respondents do not fast paediatric patients, 45% fast them for 4 hours or less, and 59% (83/141) for 6 to 12 hours before anaesthesia. Fifty-seven percent (57%, 78/137) of respondents give free access to water to paediatric patients before anaesthesia.

Ten percent (10%, 14/138) of respondents do not fast debilitated or geriatric patients, 26% (36/138) fast them for 4 hours or less, and 72% (99/138) for 6 to 12 hours before anaesthesia. Fifty-seven percent (57%, 78/138) of respondents give free access to water to debilitated or

**Table 2. Relations between risk factors.**

| Risk 1 | Risk 2 | *P*-value | Comments (*see* text for details) |
|---|---|---|---|
| Gender | Year of graduation | *0.01* | More men graduated more than 15 years ago |
| | Number of veterinarian(s) | 0.67 | |
| | On-call hours | *0.002* | More men have on-call hours activity |
| | Size of town | 0.67 | |
| | Type of practice | 0.83 | |
| | Number of animal(s) anaesthetised/day | 0.06 | |
| Year of graduation | Number of veterinarian(s) | *0.003* | More respondents, graduated less than 15 years ago, work in large team practices (5+ practitioners) |
| | On-call hours | 0.51 | |
| | Size of town | 0.07 | |
| | Type of practice | *0.001* | Respondents, graduated less than 15 years ago, more often work in referral centre |
| | Number of animal anaesthetised/ day | *<0.001* | Respondents, graduated less than 15 years ago, perform more anaesthesia cases per day |
| Number of animal(s) anaesthetised/day | Type of practice | *<0.001* | More animals are anaesthetised per day in referral centre |
| | Number of veterinarian(s) | *<0.001* | More animals are anaesthetised in large team practices (5+ veterinarians) |

geriatric patients before anaesthesia, 34% (47/138) of respondents removed water 20 min, and 12% (17/138) 6–12 hours before anaesthesia.

**Pre-anaesthetic evaluation.** Respondents answered that a complete physical examination is performed for all patients (89%, 129/145), paediatric (89%, 129/145), geriatric (99%, 144/145) or debilitated (99%, 144/145) patients in pre-anaesthetic evaluation. The examination is performed the same day of anaesthesia, both for routine surgeries (73%, 101/138) and for other surgeries (82%, 107/130).

During physical examination, respondents evaluate the following parameters: cardiac auscultation (98%, 138/141), thoracic auscultation (89%, 125/141), heart rate (87%, 123/141), respiratory rate (80%, 113/141), temperature (79%, 111/141), abdominal palpation (76%, 107/141), lymph node palpation (71%, 100/141), peripheral pulse palpation (63%, 89/141). Patient history, including appetite, drinking, urination and defecation is obtained by 84% (118/141) of respondents. Forty-three percent (43%, 61/141) of respondents evaluate all the physical parameters and obtain a history. Respondents graduated less than 15 years ago more often perform abdominal palpation than respondents graduated more than 15 years ago (76% *vs*. 61%, $P = 0.04$).

Additional diagnostic tests are recommended by 61% (84/137) of the respondents for all patients, 62% (85/137) for paediatric patients, 93% (128/137) for geriatric patients and 97% (133/137) for patients they consider at-risk. In the practitioner's perspective, these procedures are accepted by owners of patients at risk (89%, 117/132), geriatric (76%, 100/131), healthy (14%, 17/125) and a few of paediatric patients (12%, 15/122). Among the diagnostic tests, serum biochemistry (including liver enzymes, urea, creatinine and glucose) is the most frequently recommended. Respondents recommend these tests for geriatric (96%, 127/132), patients considered at-risk (95%, 126/133), paediatric (73%, 82/113) and healthy patients (67%, 82/123). Haematology is recommended for patients at risk (91%, 121/133), geriatric (84%, 111/132), healthy (33%, 41/123) and paediatric patients (33%, 37/113). Packed cell volume (PCV) and total solids (TS) are recommended for paediatric (56%, 63/113), healthy (52%, 64/123), geriatric (34%, 45/132) and patients considered at risk (34%, 45/133). Electrocardiogram (ECG) is recommended by 25% (33/135) of respondents for patients considered at-risk

and by less than 7% (9/132) for other patients. In referral centre, diagnostic tests are more often recommended for all patients (72% *vs.* 50%, *P* = 0.03), and are more commonly accepted by the owner (82% *vs.* 51%, *P* < 0.001) compared to GP conditions. The procedures more often recommended in referral centre compared to GP are: PCV/TS for healthy patients (75% *vs.* 32%, *P* < 0.001), PCV/TS (72% *vs.* 32%, *P* < 0.001) and glucose (72% *vs.* 44%, *P* = 0.005) for paediatric patients, ECG for geriatric patients (16% *vs.* 3%, *P* = 0.02) and ECG (44% *vs.* 15%, *P* < 0.001) and electrolyte measurements (75% *vs.* 51%, *P* = 0.02) for patients at-risk. Respondents in referral centre less often recommend haematology for healthy (9% *vs.* 31%, *P* = 0.01) and paediatric patients (9% *vs.* 27%, *P* = 0.04) than respondents in GP.

American Society of Anesthesiologists (ASA) physical status classification is evaluated by 35% (46/131) of respondents for routine surgery, and by 46% (59/128) for the other surgeries.

### Part III—Anaesthesia procedure

**Availability of emergency drugs.**   Overall, 39% (53/135) of respondents calculate emergency drug doses before anaesthesia for all procedures, 38% (52/135) for procedures considered at-risk and 22% (30/135) never do. Ninety-three percent (93%, 126/135) of respondents have access to an emergency crash cart, with drugs and equipment for cardiopulmonary resuscitation (CPR). Among emergency drugs, 95% (121/127) of respondents use epinephrine, 91% (117/128) atropine, 90% (111/124) glycopyrrolate, 76% (92/121) doxapram, 30% (29/97) dobutamine, 28% (27/97) dopamine, 23% (21/92) vasopressin, 22% (21/94) phenylephrine and 15% (14/94) ephedrine. Frequency of use for each drug is illustrated (see Fig 1), which shows that practices regularly use anticholinergic (atropine and glycopyrrolate) and catecholamines-like substances (dopamine and dobutamine) drugs, with the type of practice having major influence. Respondents in referral centre have more often access to phenylephrine (55% *vs.* 12%,

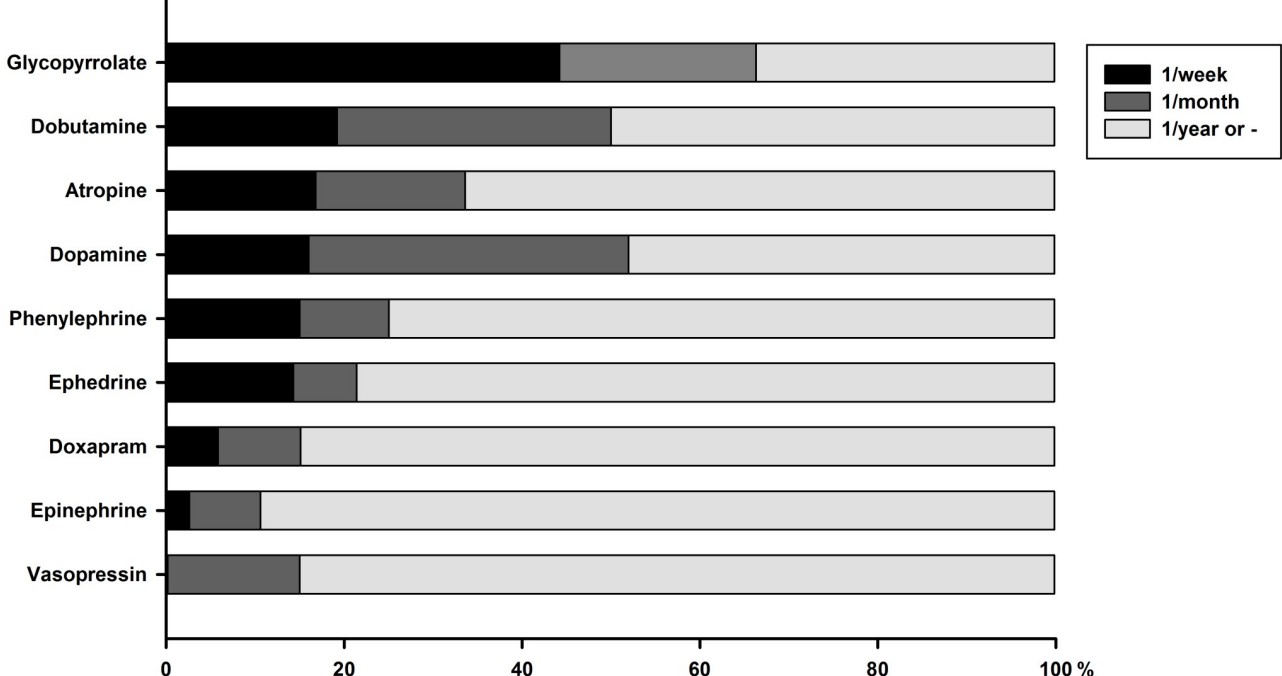

**Fig 1. Cumulative percentage of respondents reporting their frequency of use for each emergency drug in small animals anaesthesia.** Frequency of use is color-coded for at least 1/week, 1/month, and 1/year or less.

$P < 0.001$), ephedrine (38% *vs*. 8%, $P = 0.002$), dobutamine (88% *vs*. 8%, $P < 0.001$), dopamine (92% *vs*. 7%, $P < 0.001$) and vasopressin (82% and 4%, $P < 0.001$) than respondents in GP.

Among respondents using drugs that could be antagonised, 93% (124/133) report to use naloxone, 66% (68/103) atipamezole, 36% (35/98) yohimbine, 26% (24/93) flumazenil and 18% (16/88) tolazoline. Respondents graduated less than 15 years are more likely to have flumazenil (38% *vs*. 15%, $P = 0.02$). Respondents in referral centre have more often access to atipamezole (91% *vs*. 60%, $P = 0.009$) and flumazenil (91% *vs*. 7%, $P < 0.001$) than respondents in GP.

**Premedication.** Premedication is used by all respondents: 31% (40/128) use a premix (mix prepared ahead of time, same dosage for all patients), 10% (13/128) use the same protocol for all patients but mix drugs just before administration, and 59% (75/128) use individualised protocols, with different drugs and doses for each patient. The frequency of use of each drug for routine surgery is summarised in Fig 2. Briefly, non-steroidal anti-inflammatory drugs (NSAID), opioids (hydromorphone and butorphanol), acepromazine and glycopyrrolate are commonly used for routine surgeries. Two respondents (2/91) report not using opioids for routine surgeries. Veterinarians who graduated more than 15 years ago were more likely to report never using midazolam (56% *vs*. 27%, $P = 0.007$) than those who graduated more recently. For routine surgeries, GPs were more likely to report never using midazolam (52% *vs*. 0%, $P < 0.001$) and fentanyl (75% *vs*. 43%, $P = 0.005$) than respondents in referral centre. They were also more likely to highly use glycopyrrolate (43% *vs*. 5%, $P = 0.001$) and butorphanol (30% *vs*. 0%, $P = 0.03$).

The following drugs are used in premedication by respondents for non-routine surgeries: NSAID (95%, 88/93), butorphanol (90%, 88/98), hydromorphone (90%, 86/93), glycopyrrolate (84%, 77/92), atropine (84%, 77/92), acepromazine (80%, 81/101), dexmedetomidine (79%, 70/88), diazepam (77%, 68/88), buprenorphine (70%, 57/81), midazolam (63%, 55/87), fentanyl (47%, 35/75), morphine (29%, 21/73), medetomidine (25%, 19/77) and xylazine (16%,

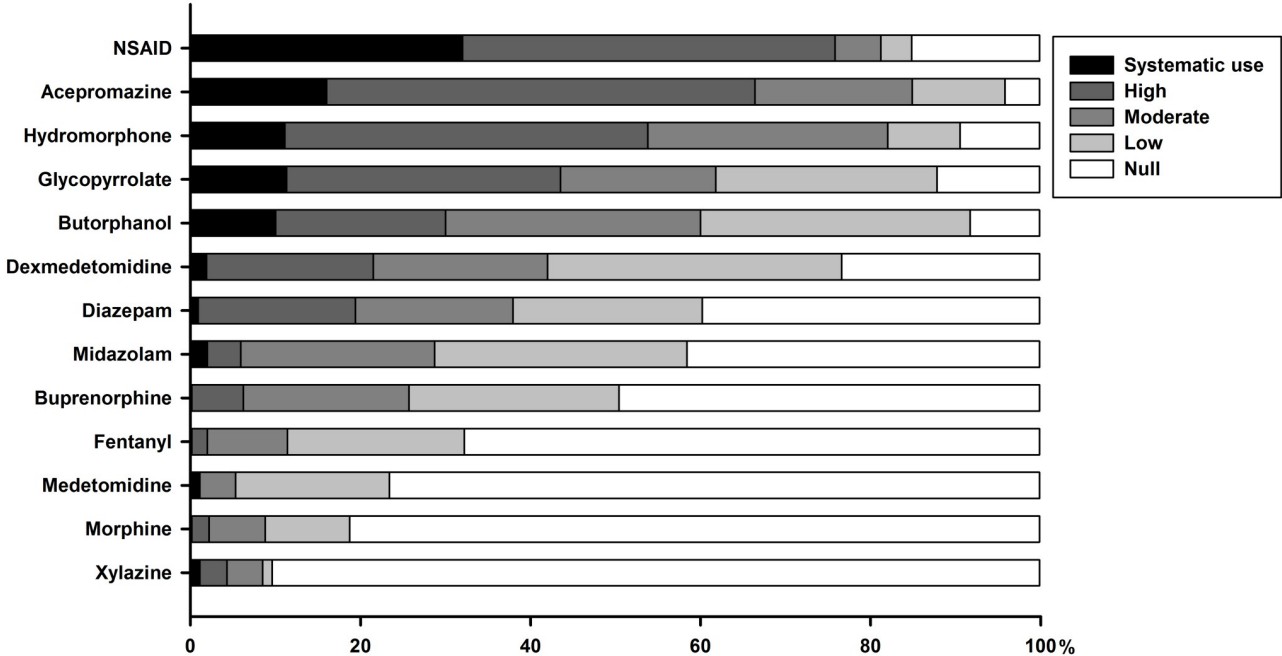

**Fig 2. Cumulative percentage of respondents reporting their frequency of use for each drug administered in small animals premedication.**
Frequency of use is color-coded, as systematic (or 100%), high (61 to 99%), moderate (21 to 60%), low (1 to 20%) or null (or 0%).

11/69). Respondents in referral centre more often use midazolam (100% *vs*. 51%, *P* < 0.001), morphine (58% *vs*. 18%, *P* = 0.002), buprenorphine (95% *vs*. 63%, *P* = 0.009), and fentanyl (90% *vs*. 31%, *P* < 0.001) than respondents in GP for non-routine surgeries.

**Induction.**   The drugs used by respondents for induction for routine surgeries are propofol (84%, 89/106), ketamine combined with diazepam (78%, 87/111), thiopental (66%, 69/104), alfaxalone (36%, 33/91) and ketamine alone (32%, 31/96). More GPs never or rarely use alfaxolone (72% *vs*. 33%, *P* = 0.03) and propofol (75% *vs*. 33%, *P* = 0.008) than respondents in referral centre. They are also more likely to use ketamine-diazepam compared to respondents in referral centre (27% *vs*. 14%, *P* = 0.01).

The drugs used by respondents for induction of non-routine surgeries are propofol (85%, 81/95), ketamine combined with diazepam (74%, 66/89), alfaxalone (45%, 33/73), thiopental (41%, 34/83) and ketamine alone (39%, 30/76). The respondents in referral centre more often use alfaxolone for non-routine surgeries compared to respondents in GP (89% *vs*. 30%, *P* < 0.001).

**Maintenance.**   Anaesthesia with injectable agents alone is performed by 36% (47/132) of respondents. Respondents in referral centre more often use this technique than respondents in GP (62% vs. 29%, P = 0.003). Drugs used for maintenance include: propofol (57%, 26/46), ketamine (44%, 20/46), a mix including ketamine, dexmedetomidine and an opioid (39%, 18/46), and alfaxalone (17%, 8/48). Respondents in referral centre more often use alfaxalone (40% *vs*. 6%, *P* = 0.01) than respondents in GP. Anaesthesia with injectable agents alone is mostly used (90%) for procedures considered rapid to perform and mildly painful by the respondents such as handling, castration of a male cat, skin biopsy, arthrocentesis or computed tomodensitometry.

When using inhalant anaesthesia, 99% (128/129) of respondents use isoflurane and 1% (1/129) use sevoflurane.

**Anaesthesia machine.**   Among respondents using inhalant anaesthesia, 95% (123/130) possess a Bain circuit (modified Mapleson D) and 94% (122/130) a rebreathing system. Six respondents (5%, 6/130) possess only a rebreathing circuit and 6 (5%, 6/130) only a Bain circuit.

**Analgesia.**   Concerning analgesia, 4% (6/147) of respondents consider that patients rarely need analgesia after surgery. Seventy-one percent (71%, 106/150) of respondents never discussed the use of analgesia with owners. Respondents in referral centre never give that choice to the owner whereas 37% of the respondents in GP do (*P* < 0.001).

All respondents use NSAID when appropriate: 63% (82/130) during recovery, 18% (23/130) at the same time as premedication, 13% (17/130) during surgery before the incision, 6% (8/130) during surgery but after the incision. After surgery, 82% (107/130) use NSAID for 3 to 4 days, 9% (12/131) during 7 days, and 9% (12/131) only administer NSAID once peri-operatively. Respondents in referral centre use more frequently a 7-day treatment than respondents in GP (21% *vs*. 6%, *P* = 0.02). If NSAIDs are used, the respondents' preferred NSAID in dogs and cats for post-anaesthetic analgesia is reported in Table 3.

Among respondents, 95% (124/130) use opioids after surgery: 11% (15/130) only administer one dose after surgery, 38% (49/130) only administer opioids as needed, 46% (60/130) administer systematically one dose after surgery and repeat as needed and 5% (6/130) never use opioid post-surgery. Respondents in referral centre use more frequently a systematic post-anaesthetic administration followed by additional doses as needed compared to respondents in GP (**75%** *vs*. 39%, *P* = 0.02). The respondents' preferred opioid in dogs and cats for post-anaesthetic analgesia is reported (Table 3). Respondents graduated less than 15 year ago more often use hydromorphone in dogs (86% *vs*. 67%, *P* = 0.03). Respondents in referral centre

**Table 3. Respondents' preferred NSAID and opioid in dogs and cats for post-surgery analgesia.**

|  | Dog | Cat |
| --- | --- | --- |
| **NSAIDs** |  |  |
| Meloxicam | **45%** | **68%** |
| Carprofen | **31%** | 1% |
| Tolfenamic acid | 8% | **24%** |
| Deracoxib | 11% | 0% |
| Firocoxib | 4% | 0% |
| Ketoprofen | 0% | 6% |
| Robenacoxib | 0% | 0% |
| **Opioids** |  |  |
| Hydromorphone | **76%** | **45%** |
| Buprenorphine | 7% | **43%** |
| Butorphanol | **12%** | 10% |
| Morphine | 4% | 3% |

The two most frequently used drugs in each species are in bold.

more often use buprenorphine in cats (79% *vs*. 35%, *P* = 0.006). Opioids and NSAID are used together by 87% (112/129) of respondents.

Sixteen percent (16%, 21/128) of respondents provide analgesia as an IV infusion during surgery. Respondents in referral centre use this technique more frequently (67% *vs*. 5%, *P* < 0.001). The drugs most frequently used are fentanyl (86%, 18/21), ketamine (67%, 14/21) and lidocaine (67%, 14/21). Respondents in referral centre use more frequently fentanyl (100% *vs*. 40%, *P* = 0.008).

Eighty-three percent (83%, 109/131) of respondents use locoregional analgesic techniques. The techniques most frequently used are ring block for declawing (89%, 97/109), maxillary (31%, 34/109), mandibular (29%, 32/109), infra-orbital (19%, 21/109), and mental (16%, 18/109) blocks. Twenty-five percent (25%, 27/109) of respondents answered performing other type of blocks, among which local splash or infiltration, and testicular block are the most frequent. Respondents in referral centre perform maxillary (52% *vs*. 26%, *P* = 0.02), mandibular (52% *vs*. 23%, *P* = 0.01), infra-orbital (35% *vs*. 15%, *P* = 0.04) and other type of blocks (52% *vs*. 17% *P* = 0.001), more frequently than respondents in GP.

## Part IV—Monitoring and safety

Technical procedures performed for anaesthesia are summarised for dogs (see Fig 3) and cats (see Fig 4). There are similarities in these anaesthetic acts both in dogs and cats, but endotracheal intubation and intravenous catheterisation are more frequent in the dog than in the cat. Systematic use of fluid therapy is infrequent, in particular in cats. Canine patients undergo endotracheal intubation, intravenous catheterisation and fluid therapy more commonly when attended by respondents graduated less than 15 year ago (*P* < 0.02). Feline patients undergo more commonly intravenous catheterisation by respondents graduated less than 15 year ago (33% *vs*. 17%, *P* = 0.03). Type of practice also impacts how systematic are those procedures.

When performing anaesthesia with injectable drugs only, respondents provide oxygen to the patient using a mask (38%, 38/100), using endotracheal intubation connected to an anaesthetic machine (33%, 33/100), by placing the oxygen in front of the patient nose (3%, 3/100) and 26% (26/100) do not provide oxygen to the patient.

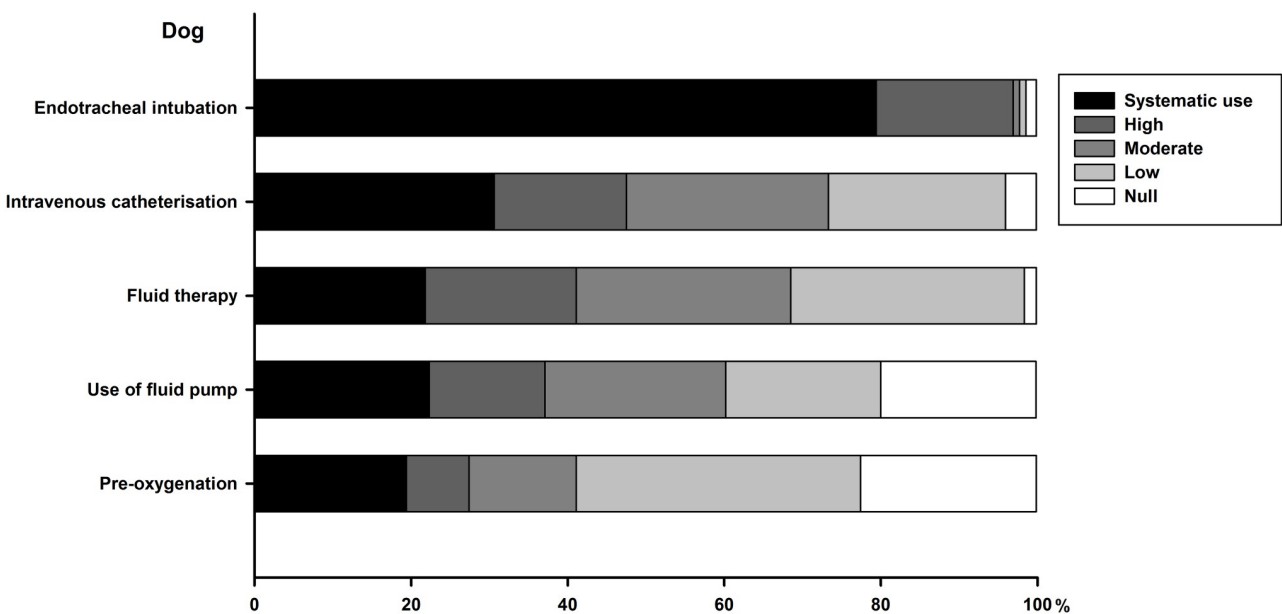

**Fig 3. Cumulative percentage of respondents reporting their frequency of use for each technical procedure performed for dog anaesthesia.**
Frequency of use is color-coded, as systematic (or 100%), high (61 to 99%), moderate (21 to 60%), low (1 to 20%) or null (or 0%).

When evaluating the cardiovascular function, respondents report to monitor heart rate (96%, 125/130), mucous membrane colour and capillary refill time (81%, 105/130), systemic arterial blood pressure (57%, 74/130), cardiac auscultation (40%, 52/130), peripheral pulse (38%, 49/130) and ECG (27%, 35/130). Respondents graduated less than 15 years ago more often use ECG (35% *vs*. 18%, *P* = 0.03). Respondents in referral centre more often monitor

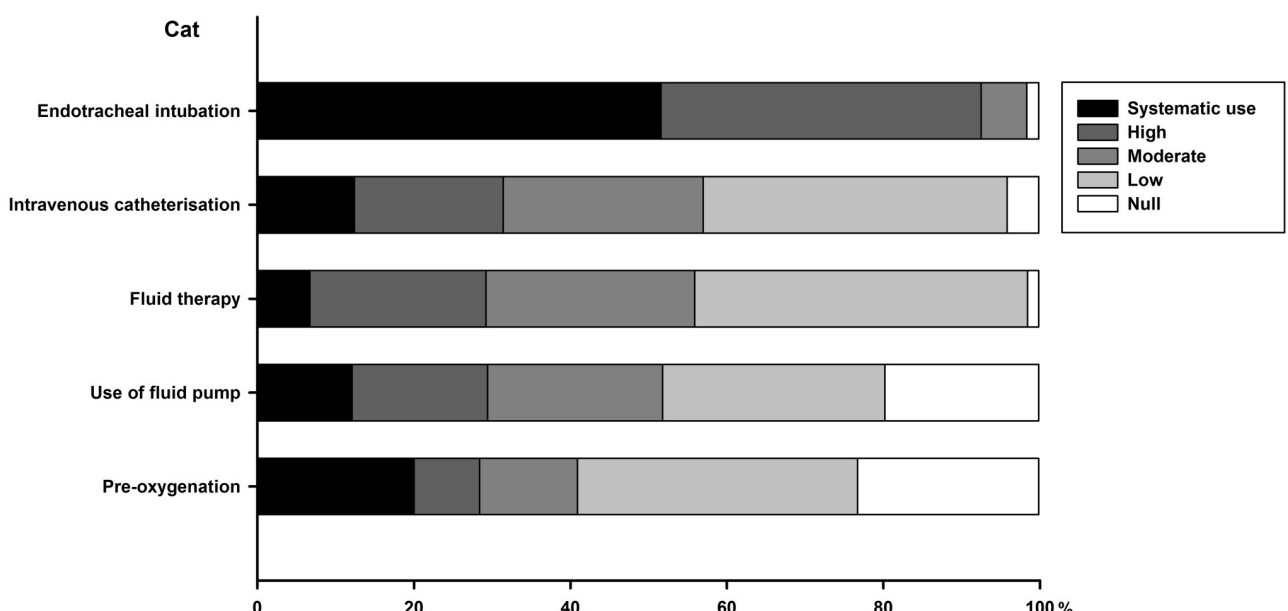

**Fig 4. Cumulative percentage of respondents reporting their frequency of use for each technical procedure performed for cat anaesthesia.**
Frequency of use is color-coded, as systematic (or 100%), high (61 to 99%), moderate (21 to 60%), low (1 to 20%) or null (or 0%).

ECG (58% *vs.* 19%, *P* < 0.001) and systemic arterial blood pressure (96% *vs.* 49%, *P* < 0.001) than respondents in GP.

For the respiratory function, respondents reported to monitor respiratory rate (91%, 119/131), pulse oximetry (89%, 117/131), capnography (25%, 33/131) and lung auscultation (24%, 31/131). Fifty-one percent (51%, 67/131) of respondents use a device to monitor the respiratory rate. One respondent uses blood gas analysis (1/131) to monitor cardio-respiratory function. Respondents who graduated more than 15 years ago more often monitor respiratory rate (32% *vs.* 16%, *P* = 0.04). Respondents from referral centre more often use capnography (71% *vs.* 15%, *P* < 0.001).

For the neurological function, respondents monitor palpebral reflex (94%, 120/127), jaw tone (60%, 76/127), pharyngeal reflex (51%, 65/127), eye position (48%, 61/127) and withdrawal reflex (32%, 41/127). Respondents who graduated less than 15 years ago monitor eye position more frequently (59% *vs.* 39%, *P* = 0.03) than respondents graduated more than 15 years ago. Respondents in referral centre more often monitor jaw tone (92% *vs.* 52%, *P* < 0.001) and eye position (71% *vs.* 43%, *P* = 0.02) than respondents in GP.

Availability and use of monitoring devices by respondents are reported (Table 4). Respondents in referral centre have more often access to ECG (86% *vs.* 44%, *P* < 0.001), oesophageal stethoscope (43% *vs.* 21%, *P* = 0.03), capnography (82% *vs.* 18%, *P* < 0.001), blood gas analyser (75% *vs.* 3%, *P* < 0.001), oscillometric (54% *vs.* 30%, *P* = 0.03) and invasive systemic arterial blood pressure (61% *vs.* 1%, *P* < 0.001).

When needed, complementary exams can be performed during the procedure by 93% (120/129) of the respondents. Respondents graduated less than 15 years ago have more often access to blood gas analysis (33% *vs.* 13%, *P* = 0.01), electrolytes (67% *vs.* 44%, *P* = 0.02), blood type (32% *vs.* 10%, *P* = 0.004) and crossmatch (35% *vs.* 13%, *P* = 0.006). Respondents in referral centre have more often access to complete blood count (100% *vs.* 61%, *P* < 0.001), blood chemistry (100% *vs.* 75%, *P* = 0.008), blood gas analysis (100% *vs.* 4%, *P* < 0.001), electrolytes (100% *vs.* 44%, *P* < 0.001), blood type (96% *vs.* 2%, *P* < 0.001) and crossmatch (100% *vs.* 5%, *P* < 0.001).

Transfusion is not an option for 68% (88/129) of the respondents. Respondents in referral centre have more often access to blood transfusion (92% *vs.* 18%, *P* < 0.001) than respondents in GP.

A ventilator is available for 25% (32/129) of respondents. Respondents graduated less than 15 years ago have more often access to a ventilator (35% *vs.* 15%, *P* = 0.01). However, fewer respondents graduated less than 15 years ago use it routinely (5% *vs.* 60%, *P* = 0.001) compared with respondents graduated more than 15 years ago. Respondents in referral centre have more

**Table 4. Availability and use of monitoring devices by respondents.**

| Monitoring device | Present in the clinic | Used in routine cases (%) | Used in non-routine cases (%) | Present but not used (%) |
|---|---|---|---|---|
| Pulse oximeter | 123 | **90%** | 72% | 3% |
| Doppler blood pressure measurement | 90 | 50% | **70%** | 18% |
| ECG | 74 | 35% | 77% | 18% |
| Oscillometric blood pressure measurement | **50** | 44% | **66%** | 24% |
| Multi-parametric monitor | **48** | **100%** | 75% | **0%** |
| Capnograph/Capnometer | **43** | 58% | **81%** | 7% |
| Apnoea monitor | 42 | 67% | 62% | 19% |
| Oesophageal stethoscope | 36 | 22% | 47% | 50% |
| Blood gases analyser | **24** | 0% | **87%** | 12% |
| Invasive blood pressure measurement | **18** | 11% | **83%** | 11% |

often access to a ventilator (92% *vs*. 10%, *P* < 0.001) and are more likely to use if they suspect they would need it (100% *vs*. 70%, *P* = 0.02).

During routine surgeries, monitoring is performed by someone dedicated to this task (24%, 31/131), someone helping with the surgery (56%, 74/131) or the person doing the surgery (20%, 26/131). During non-routine surgeries, monitoring is performed by someone dedicated to this task (40%, 50/126), someone helping with the surgery (54%, 68/126) or the person doing the surgery (6%, 8/126). Referral centres are more likely to have someone dedicated to monitoring for both routine (46% *vs*. 18%, *P* = 0.01) and non-routine surgeries (79% *vs*. 30%, *P* < 0.001).

Monitoring data are systematically recorded on an anaesthesia record by 29% (38/130) of the respondents, sometimes by 16% (20/130), and never by 55% (72/130). More respondents in GP never record monitoring data (67% *vs*. 4%, *P* < 0.001) than respondents from referral centre. Recording is reported to be performed every 5 min (29%, 38/131), every 10 minutes (14%, 18/131), every 15 minutes (1%, 2/131), only at the beginning and the end of the anaesthesia (1%, 2/131) or at no specific interval (55%, 72/131, of respondents). Respondents in GP are more likely to record monitoring data at no specific interval than respondents in referral centre (64% *vs*. 0%, *P* < 0.001). When monitoring data are recorded, the most frequently included parameters are heart rate (96%, 22/23), respiratory rate (74%, 17/23), systemic arterial blood pressure (70%, 16/23), temperature (61%, 14/23), oxygen saturation (43%, 10/23) and end-tidal carbon dioxide (43%). Respondents in referral centre more often record blood pressure (55% *vs*. 17%, *P* = 0.006).

During recovery, monitoring of the patient include visual monitoring (eye position, mucous membrane colour, thoracic movements—93%, 121/130), temperature (84%, 109/130), tactile monitoring (pulse quality, jaw tone, palpebral reflex—75%, 98/130), auscultation (68%, 88/130) and the same monitoring as during anaesthesia (11%, 14/130). Respondents graduated less than 15 years ago are more likely to monitor their patient temperature during recovery (88% *vs*. 73%, *P* = 0.03). Respondents in referral centre are more likely to palpate their patient during recovery (96% *vs*. 67%, *P* = 0.006).

Monitoring during recovery is continued until the patient is able to remain in sternal recumbency (58%, 76/130), the patient temperature is considered normal (50%, 65/130) and/or the patient is extubated (41%, 53/130). Respondents graduated less than 15 years ago are more likely to continue the monitoring until the patient temperature is normal (61% *vs*. 37%, *P* = 0.006) but less likely to continue it until the patient is able to stay sternal (47% *vs*. 65%, *P* = 0.04). For routine surgery, respondents stop rewarming the patient when its rectal temperature reaches 36˚C (5%, 6/128), 37˚C (**58%**, 74/128) or 38˚C (16%, 20/128). Twenty two percent (22%, 28/128) of the respondents do not always measure temperature during recovery. More respondents in GP do not measure their patient temperature compared to respondents in referral centre (27% *vs*. 0%, *P* = 0.007).

After routine surgery, 51% (66/130) respondents keep the patient hospitalised between 12 to 24 hours after surgery, 31% (40/130) for 6 to 12h after surgery, 15% (19/130) more than 24h after surgery and 4% (5/130) less than 6h after surgery.

## Discussion

Our objective was to report how small animals anaesthesia is performed in veterinary practices in Quebec. Interestingly, discrepancies exist between our observations and the guidelines for anaesthesia and analgesia that have been published [12–16].

Veterinary practices in Quebec do not follow the guidelines published by the American Animal Hospital Association (AAHA) [14], specifically regarding client management, access

to water before anaesthesia, fasting paediatric patients, patient evaluation, physical examination, use of the ASA physical status classification, preparation for emergency, use of individualised anaesthesia plan, analgesia procedure(s), endotracheal intubation, intravenous catheterisation, use of anaesthesia record, and monitoring during anaesthesia and recovery. Similarly, the American College of Veterinary Anesthesia and Analgesia (ACVAA) published guidelines for monitoring during anaesthesia [12]. Based on these guidelines, half of the respondents perform cardiovascular monitoring as recommended, half the monitoring of oxygenation and 25% the monitoring of ventilation. Record keeping and personnel attending the patient during anaesthesia do not meet the ACVAA standard for about 50 to 75% of the respondents, when only 23 respondents described their anaesthesia monitoring outcomes. The use of analgesia does not meet the requirements of the AAHA [16], or of the World Small Animal Veterinary Association (WSAVA) [15], especially regarding patients that have surgery without analgesia because the choice of pain relief was given to the owner. Overall, the way anaesthesia is performed by most respondents does not meet the published criteria, at the time of the study. Additionally, guidelines published more recently [13, 18] tend to be similar to those published earlier to which the respondents still do not meet.

The evaluation and preparation of the patient appears to be sub-optimal in many instances in this study. Recommendation regarding fasting is to withhold food at least 6h prior to anaesthesia [11]. Water can be allowed until just prior to anaesthesia. Dogs and cats less than 8 weeks old should not be fasted for more than 2h [11]. Based on these recommendations, only 2% of the respondents do not meet the criteria for food withholding in adult patients and 49% for water. The percentage of respondents who do not meet the criteria is higher for paediatric patients, with 43% fasted for more than 4 hours and not given free access to water. Therefore, it seems most practices do not follow the guidelines in term of pre-anaesthesia fasting, except for food withholding for adult patients. However, it is worth mentioning that the guidelines have changed over the past decades. It was previously recommended to allow free access to water until up to 2 hours [19], 2–4 hours [20], or at least 2 hours before anaesthesia [21]. This highlights that some practices may still follow the older guidelines and failed to update their standards.

Guidelines regarding provision of an informed consent are well followed. Providing additional information about anaesthesia and its related risks, or pain management is not so widely done despite Canadian [22] and American [23] pet owners' documented concerns.

Assessing health status of patients is an important part of the risk management during small animals anaesthesia. In our study, only 43% of respondents evaluate all the physical parameters and obtain a history. Only one third of the respondents evaluate ASA physical status grade. Patients with increased ASA grade have higher risk of death [1–10]. Pre-existing diseases can interfere with the pharmacology of drugs used during anaesthesia and decrease the ability to cope with drug side-effects. Even if there is no consensus as to the optimal method of patient health assessment, guidelines agree that the physical health should be thoroughly assessed [11, 13, 14]. Results of the survey suggest that Quebec practitioners pay attention to the cardio-respiratory system. Pre-anaesthetic blood testing is inconsistent between respondents. There is currently no consensus for their use in healthy patients undergoing elective surgery [11]. In a study of 101 geriatric dogs (older than 7 years of age), standard physical evaluation and history confirmed 13 pre-existing conditions, and routine serum chemistry and haematology detected 30 new conditions based on history and physical examination [24]. This demonstrates that pre-anaesthetic blood testing in patients that are geriatric or considered at risk is useful and recommended.

Pain management is vital for all patients undergoing surgery. Despite guidelines [15, 16] and legal obligation (Loi sur la protection sanitaire des animaux 1993, c. 18, s. 6; 2000, c. 40, s.

29.), 29% of respondents (and even 37% in GP) still present analgesia as a choice of clients for patients undergoing routine surgery. Opioids, despite being considered the cornerstone of effective pain management, is never used by 5% of respondents. Even though most respondents treat pain, the number of patients potentially not receiving analgesia is cause of concern. In particular, butorphanol remains commonly used, despite demonstration of its poor to limited analgesic effect [25–27]. This is inadequate in light of the discovery of a very low expression of OPRK1, the gene for kappa opioid receptor, in the dorsal root ganglion of the dog (0.01 significant fragments per kilobase per million aligned reads–sFPKM–), compared to the rat (3.19 sFPKM) and human (3.48 sFPKM) [28]. This explains why butorphanol, as a kappa-agonist, could be effective in some species, but not in dogs. Additionally, premixes are commonly used even though they prevent adjusting anaesthesia and analgesia to the patient specific needs. Finally, opioid concentration in premedication premixes may be too low to provide sufficient analgesia. Constant analgesia drug infusions are not popular (17%). Seventeen percent of respondents (17%) do not perform any local block. While ring blocks for declawing are widely used (89% of those performing this surgery), all other loco-regional blocks are seldom practiced (25 to 30%). As those techniques are inexpensive and easy to learn and implement, pain management could be easily improved with low relative risk (15, 16).

There is a wide disparity between practices in the way anaesthesia is practiced. Some practices have no or limited access to emergency drugs, antagonists, vasopressors and inotropes. More than 40% of practices do not use individualised premedication protocols. A few practices do not have access to an anaesthesia machine suitable to most small animal patients. Endotracheal intubation, intravenous catheterisation and fluid therapy are not routinely performed, particularly in cats. Therefore, patient safety could be a concern, especially when there is limited access to drugs used to treat anaesthesia complications, or when drugs are not tailored to the patient's need. Endotracheal intubation protects airways and allows administration of oxygen. Placing an intravenous catheter provides venous access for administration of emergency drugs. Intravenous administration of fluids is not widely used, even though it participates in the management of fluid balance and facilitates the elimination of anaesthetic drugs, both of which are important during short or long procedure [13, 14, 18, 29], especially considering the withholding time of food and water reported in this survey.

Regarding monitoring, a majority of practices do not use any anaesthetic record and monitoring equipment available during anaesthesia varies between practices. Guidelines [13, 14, 18] and textbooks [11, 29] recommend using individualised protocols and monitoring, in order to decrease the veterinarian liability in case of litigation. Additionally, the Ordre des Médecins Vétérinaires du Québec requires the systematic use of an anaesthesia record in Quebec practice. Monitoring the arterial pulse and the use of pulse oximetry have been linked with reduction in odds of anaesthetic-related deaths in veterinary medicine [7]. Odds of death for cats were greater when preanesthetic physical exam and oxygen saturation were not recorded [30]. In human medicine, severe hypotension (defined as a more than 40 or 50% decrease in systemic arterial blood pressure relative to each patient's baseline, lasting more than 5 minutes) is associated with acute kidney injury (27% occurrence, with a more than doubled risk with an intraoperative reduction in systemic arterial blood pressure more than 50%) [31] and myocardial damage [32].

It is worth noting that not all practices are the same. Specifically, this study highlights some differences between GP and referral centre. Overall, referral centres communicate better, do not expose patients to surgically-induced pain, have better access to drugs and equipment, recommend more additional diagnostic tests and perform a closer monitoring with the assistance of more support tools and staff. The impact of professional interaction involving specialists, service providers, veterinarians and technicians, its difference between GP and referral centre,

were not evaluated in the present study. Mortality and morbidity rates were beyond the scope of this study. Therefore, the relationship between level of care and outcome could not be explored. The lack of monitoring, specifically, could put patient at a higher risk. Based on this study, the main differences between GPs and referral centres regarding monitoring are the access and use of ECG, blood pressure monitor, capnography, and measurement of temperature. Referral centres are also more likely to have a person dedicated to anaesthesia during routine or non-routine surgeries, and to have more ready access to equipment to manage complications should they arise. Considering that the reported risks of arrhythmias (2.5–4% in dogs [3, 4, 33] and 1.8–3.6% in cats [3, 4, 33], hypotension (7 to 63% [33–35] in dogs, 8.5% in cats [33]), hypoventilation (1.3% to 60% [33–34] in dogs, 1/683 in cats [33]), and hypothermia (up to 92% in dogs [36], 98% in cats [37]) are significant, and considering that it has already been reported that having a nurse monitoring the anaesthesia decreases the mortality risk [2], patients undergoing anaesthesia in referral centres may be less at risk as access and use of monitoring devices are more frequent than in GPs. Anxiety about anaesthetising dogs with heart disease is a common cause of referral to speciality centres. It is interesting to note that dogs with heart disease (n = 100), when anaesthetised by trained personnel and carefully monitored during routine dental procedures (in a teaching hospital), were not at significantly increased risk for anaesthetic complications [38]. Considering that such additional tools and procedures require a considerable investment, it would be interesting to compare referral centre and GP conditions for morbidity and mortality in similar anaesthetic procedures, remembering that even a rare anaesthesia-related death has a marked impact on clients and the veterinary staff. This would provide fundamental knowledge for guiding developments in veterinary and continuing education.

The demographic results revealed some interesting associations. Not surprisingly, a clear gender effect was associated with the year of graduation, with more men present in older graduates and more women in younger graduates. This is consistent with current veterinary school enrolment data that shows that more than 81% of students enrolled in veterinary school of AVMA-accredited colleges (n = 46) are female [39]. A minority of respondents (18.6%) practice on-call hours emergency, and in ratio a majority of men is doing it. Finally, younger graduates more often work in referral centres. Year of graduation impacts significantly some of the responses. However, overall, year of graduation does not influence the standard of care reported in this study, except on safety aspects (intravenous catheterisation, endotracheal intubation, fluid therapy). We selected this limit of 15 years, based on the data distribution for getting significant groups. It has been previously reported that recently graduated veterinarian competency is influenced by their colleagues, in particular during their first year of practice [40]. Additionally, continuing education is mandatory for any veterinarian to maintain registration in Quebec. Therefore, the differences observed between the respondents could mostly be explained by difference in availability of equipment or drugs, in-place rules, rather than difference in training. Additionally, this also highlights the importance of continuing education, and the necessity to measure the impact of academic education evolution in any discipline. Finally, it is interesting to note that gender did not influence anaesthesia procedure and pain management, such as reflected in the survey (with all its limitations: voluntary response on an electronic survey, limited power of analysis on a localised population, . . .).

This study has some limitations, in particular whether it is representative of the population surveyed and the accuracy of the observations. The form of survey may have generated a positive bias because people with a strong interest in anaesthesia may have been more likely to participate. Results of the overall veterinarian population could be different. Finally, using a company client database as the population surveyed may have introduced a sampling bias. However, the company selected is one of the two major suppliers of veterinary anaesthetic

equipment and interacts with equipment in most practices all over the Quebec province. Additionally, based on the Ordre des médecins vétérinaires du Québec (https://www.omvq.qc.ca/la-profession/profil-medecins-veterinaires.html; accessed the 08 of April, 2019), the population surveyed mirrors the veterinary population in Quebec in terms of gender, year of graduation and type of practice. Therefore, the population surveyed is likely to be representative of the veterinary population who routinely performs anaesthesia in Quebec.

In conclusion, while this study has some limitations, the results demonstrated the discrepancies existing between the standard of practice recommended by international guidelines and the level of care performed in veterinary practices in Quebec. Anaesthesia and analgesia practice in referral centre looks close to the standards promulgated in academic environment. Why does-it not look to be the same in GP? Marked differences are obvious between referral centre (and supposedly academics) and GP on client management, patient evaluation and preparation, use of the ASA physical status classification, preparation for emergency, use of individualised anaesthesia plan (31% always use a premix), analgesia procedure(s) (37% of GPs present analgesia as an option to clients, butorphanol remains quite popular in GP, whereas optimal analgesia regimen including constant rate infusion and loco-regional analgesia remain confined to a minority), safety procedures and monitoring during anaesthesia and recovery. As the year of graduation does not affect (or minimally, on safety aspect) the standard of care, is the exposure to field practice diluting the clinical skills and knowledge acquired during veterinary training? Is-it the same for other disciplines? To explain such stagnation in standard of care, it could be hypothesised that the success rate in anaesthesia/ analgesia remains unfortunately related to mortality and does not involve morbidity and animal welfare. Is the situation unique to Quebec? The poor integration of guidelines promoted by international organisations (AAHA, AAFP, ACVAA, WSAVA, *etc.*) questions if the language could be a source of limited dissemination. A comparison between referral centre and GP conditions for morbidity and mortality in similar anaesthetic procedures would provide some elements of response, as well as the comparison of this survey results to those got in the Rest of Canada.

## Supporting information

**S1 Appendix. Questionnaire.** Presentation, in English, of the questionaire used for the electronic survey, with the different sections, and all questions.
(DOCX)

**S1 File. Data responses to Q8–Q27.** Database of the responses collected from the 156 responders to the survey for the part–Evaluation and management of anaesthetic risk (see S1 Appendix for detailed questions).
(XLSX)

**S2 File. Data responses to Q28–Q45.** Database of the responses collected from the 156 responders to the survey for the part–Anaesthesia procedure (see S1 Appendix for detailed questions).
(XLSX)

**S3 File. Data responses to Q46–Q66.** Database of the responses collected from the 156 responders to the survey for the part–Monitoring (see S1 Appendix for detailed questions).
(XLSX)

## Acknowledgments

The authors wish to thank Mrs Mélissa Lachapelle (Dispomed Inc.) for her active contribution to the success of the survey.

## Author Contributions

**Conceptualization:** Geoffrey Truchetti, Colombe Otis, Eric Troncy.

**Data curation:** Guy Beauchamp.

**Formal analysis:** Geoffrey Truchetti, Colombe Otis, Anne-Claire Brisville, Guy Beauchamp, Eric Troncy.

**Funding acquisition:** Eric Troncy.

**Investigation:** Geoffrey Truchetti, Colombe Otis.

**Methodology:** Colombe Otis, Guy Beauchamp, Daniel Pang, Eric Troncy.

**Project administration:** Colombe Otis, Eric Troncy.

**Resources:** Colombe Otis, Eric Troncy.

**Supervision:** Daniel Pang, Eric Troncy.

**Validation:** Colombe Otis, Anne-Claire Brisville, Daniel Pang, Eric Troncy.

**Writing – original draft:** Geoffrey Truchetti, Colombe Otis, Anne-Claire Brisville, Guy Beauchamp, Daniel Pang, Eric Troncy.

**Writing – review & editing:** Geoffrey Truchetti, Colombe Otis, Anne-Claire Brisville, Guy Beauchamp, Daniel Pang, Eric Troncy.

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
