## [Decision Letter · Decision Letter 0]

16 Sep 2019

PONE-D-19-16258

Management of veterinary anaesthesia: A survey of current practice in Quebec

PLOS ONE

Dear Dr Eric Tronci,

Thank you for submitting your manuscript to PLOS ONE. After careful consideration, we feel that it has merit but does not fully meet PLOS ONE’s publication criteria as it currently stands. Therefore, we invite you to submit a revised version of the manuscript that addresses the points raised during the review process.

The authors should provide an esaustive response to the critiques reported by the reviwer

We would appreciate receiving your revised manuscript by Oct 31 2019 11:59PM. To enhance the reproducibility of your results, we recommend that if applicable you deposit your laboratory protocols in protocols.io, where a protocol can be assigned its own identifier (DOI) such that it can be cited independently in the future. For instructions see: http://journals.plos.org/plosone/s/submission-guidelines#loc-laboratory-protocols

We look forward to receiving your revised manuscript.

Kind regards,

Francesco Staffieri

Academic Editor

PLOS ONE

Journal Requirements:

4. In your Methods, please state where the participants were recruited for your study.

5. Thank you for stating the following in the Financial Disclosure section:

There was not proprietary interest or funding directly provided for this project. This work was indirectly supported (ETR) by a Discovery grant (#441651–2013, supporting salaries) and a Collaborative Research and Development grant (#RDCPJ 491953–2016 supporting operations and salaries in partnership with ArthroLab Inc.) from the Natural Sciences and Engineering Research Council of Canada. COT is a recipient of a MITACS Canada Elevation postdoctoral scholarship (#IT11643). The funders had no role in study design, data collection and analysis, decision to publish, or preparation of the manuscript. The specific roles of all authors are articulated in the ‘author contributions’ section.

We note that you received funding from a commercial source: ArthroLab Inc.

Additional Editor Comments:

The authors should provide an esaustive response to the critiques reported by the reviwer

Reviewers' comments:

Reviewer's Responses to Questions

**Comments to the Author**

1. Is the manuscript technically sound, and do the data support the conclusions?

Reviewer #1: Partly

2. Has the statistical analysis been performed appropriately and rigorously? 

Reviewer #1: I Don't Know

3. Have the authors made all data underlying the findings in their manuscript fully available?

Reviewer #1: Yes

4. Is the manuscript presented in an intelligible fashion and written in standard English?

Reviewer #1: Yes

5. Review Comments to the Author

Reviewer #1: Thank you for the opportunity to review this interesting manuscript. As a whole, I think it can provide interesting insight into the practice of small animal anaesthesia however I think the authors need to better explain why they feel that practice in their subsect (French Eastern Canada) might differ from elsewhere and, hence, why the study was conducted. In doing this, the end of the discussion also needs a bit better of a conclusion about what the reader should do with this information. The authors need to provide some statement about what comes next or how to fix the problem. In addition, as this was only a study of anaesthesia practice in small animal settings, I believe that needs to be made more clear. I suggest adding "small animal" to the title and including it at various places throughout the text body as well. The is critical as anaesthesia in equine, ruminant, exotic or zoo practice might differ profoundly. Also, I have some concerns about whether this study was powered sufficiently, given the relatively low survey response rate, to elucidate differences if they existed. No mention of a power study was included. Finally, I've done some minor editing however I would recommend that the manuscript be reviewed for English grammar as some of the wording is awkward.

As the original survey and subsequent data included in the supplementary attachments was in French, I was unable to review them for validity and appropriate representation within this manuscript.

The remainder of my comments will be done by referencing line numbers in the text.

Line 22: consisted OF four parts, not "in"

Line 24 and throughout the text: Year and Type do not need to be capitalised

Line 30: insert the word "prior" between evaluation and to

Line 33: Avoid emotive language. Change to "Some practices are more compliant..."

Line 39: Insert "small animal"

Line 41: Lumb and Jones is an inappropriate reference for this statement. Please reference the original papers here (you reference them later anyhow) and move the reference to the end of the sentence after the sick patients as that is part of the reference.

Line 42: Using "should also" implies that you've previously stated something else to evaluate. Delete this.

Line 43: Again, more appropriate references should be used here (Hosgood and Scholl for example).

Line 45 and other places: The WSAVA and AAHA/AAFP guidelines do not specifically make best practice recommendations about anaesthesia so this is a bit misleading. Consider changing your discussion around these references to refer more specifically to peri-operative analgesia. Also, this should include Warne et al 2018 (AVJ 96(11)) in the references.

Line 47: Not sure what is meant by "nature of professional interaction"? This statement also probably should be referenced.

Line 62: Please describe the validation process (pilot survey with focus group?).

Line 65: Should read "fell under the Article"

Table 2: "large team practices" is never defined

Line 117: "on the subsequent responses"

Line 133: How were paediatric patients defined?

Line 169: Replace "easily" with commonly

Line 177: ASA Status needs a reference and it is never defined or explained anywhere prior to this point. Please add it to the introduction.

Emergency drugs section: I would question whether monitoring status, monitoring BP in particular, had an influence on these results. Have you done any regression analysis on potential confounders?

Lines 212 and others: Reporting in the negative is odd and a bit hard for the reader to interpret. For instance, stating that fewer respondents who were more recent grads never use midazolam than older grads would make a bit more sense rewritten as "Veterinarians who graduated more than 15 years ago were more likely to report never using midazolam than those who graduated more recently." Please consider changing all instances of negative reporting for clarity.

Lines 214-216: As the survey is in French, I can't clarify but were the drug questions specifically about the premed period or could they have been interpreted as "at any time"? Also, was "routine surgeries" defined for participants? I ask because I work in referral practice and I would rarely use either midazolam or fentanyl in cases that I defined as "routine".

Line 228 and others: Should read "referral centre more often use"....this displacement of the verb occurs multiple times within the text.

Line 262: Please rewrite the line "all respondents use NSAIDs as, I would hope, it is misleading. Certainly they don't all use them on every case which is how I read this. Consider "All respondents use NSAIDs when appropriate" or something along those lines.

Line 303: Please change the word "benefit" as this study did not evaluate outcomes.

Line 363: May need to change the end of this as many practices weren't monitoring ventilation so how would they "know if they needed it"?

Lines 378-382: Should probably highlight more clearly this super low response rate somewhere in the discussion.

Line 407: Need to provide references here again.

Line 425: Please change to "The evaluation and preparation of the patient appears to be sub-optimal in many instances in this study."

Line 429: Please rewrite "Situation is even worse for...." to avoid emotive language.

Line 433: Please change reminding to mentioning.

Entire fasting section: What about Savvas et al 2009, VAA 36(6)?

Line 437: Should read "Guidelines regarding provision of an informed consent are well followed." Delete the part from "and so" as this doesn't add anything to the discussion and you don't unpack it to consider why further.

Line 440: Missing the word "documented" before concerns.

Line 450-455: Flip the order of these sentences so that the initial first sentence doesn't read like an opinion sentence.

Line 458: Is this specific for post-op or any analgesia at all?

Lines 462-465: Rewrite as "This is inadequate in light of the discovery of a very low expression....."

Lines 465-466: Add "the" before dog and rat.

Line 467: Should be "why" not "that" and "effective" not "efficient"

Line 470: Change proper to "sufficient"

Line 472: Change "If" to "While"

Line 474: Consider adding a referenced statement about low relative risk with loco regional anaesthesia.

Line 486: I think more recent fluid guidelines would be more appropriate here. Consider 2013 AAHA/AAFP guidelines.

Line 496: Define "severe" hypotension

Line 500: Is there necessary pain?

Line 510: Readily should be "ready"

Lines 511-514: Please add data from Carter et al, 2017 JAAHA 53(4)

Line 518: Change reference to referral

Line 521: Chance undergraduate to "veterinary" as many programmes are NOT undergraduate.

Lines 523: The demographic results revealed some interesting associations. Not surprisingly, a clear gender effect was associated with the year of.....

Line 525: Add "This is consistent with current veterinary school enrolment data that shows that approximately 80% of students enrolled in veterinary school are female. (AAVMC Annual Report 2018)"

Line 527: However, overall, year of graduation does not influence the standard of care reported in this study.

Line 535: What is meant be "formation"?

Line 536-538: I highly doubt this study was powered sufficiently to make this claim.

Line 542: Remove "possibly worse"

6. PLOS authors have the option to publish the peer review history of their article (what does this mean?). If published, this will include your full peer review and any attached files.

Reviewer #1: No

---

## [Author Response · Author response to Decision Letter 0]

15 Nov 2019

Please, refer to the file named Responses to the Reviewer

---

## [Decision Letter · Decision Letter 1]

16 Dec 2019

Management of veterinary anaesthesia in small animals: A survey of current practice in Quebec

PONE-D-19-16258R1

Dear Dr. Eric Troncy,

We are pleased to inform you that your manuscript has been judged scientifically suitable for publication and will be formally accepted for publication once it complies with all outstanding technical requirements.

With kind regards,

Francesco Staffieri

Academic Editor

PLOS ONE

Additional Editor Comments (optional):

Reviewers' comments:

Reviewer's Responses to Questions

**Comments to the Author**

1. If the authors have adequately addressed your comments raised in a previous round of review and you feel that this manuscript is now acceptable for publication, you may indicate that here to bypass the “Comments to the Author” section, enter your conflict of interest statement in the “Confidential to Editor” section, and submit your "Accept" recommendation.

Reviewer #1: All comments have been addressed

2. Is the manuscript technically sound, and do the data support the conclusions?

Reviewer #1: Yes

3. Has the statistical analysis been performed appropriately and rigorously? 

Reviewer #1: Yes

4. Have the authors made all data underlying the findings in their manuscript fully available?

Reviewer #1: Yes

5. Is the manuscript presented in an intelligible fashion and written in standard English?

Reviewer #1: Yes

6. Review Comments to the Author

Reviewer #1: Thank you for addressing my original concerns and suggestions. I believe the changes have improved the manuscript impact and that appropriate heed has been placed with regards to potential lack of power due to surgery response rate.

7. PLOS authors have the option to publish the peer review history of their article (what does this mean?). If published, this will include your full peer review and any attached files.

Reviewer #1: No

---

## [Editor Report · Acceptance letter]

8 Jan 2020

PONE-D-19-16258R1 

Management of veterinary anaesthesia in small animals: A survey of current practice in Quebec 

Dear Dr. Troncy:

I am pleased to inform you that your manuscript has been deemed suitable for publication in PLOS ONE. Congratulations! Your manuscript is now with our production department. 

With kind regards,

on behalf of

Dr. Francesco Staffieri 

Academic Editor

PLOS ONE